# Adaptation and Validation of a French Version of the Vaccination Attitudes Examination (VAX) Scale

**DOI:** 10.3390/vaccines11051001

**Published:** 2023-05-19

**Authors:** Margot Eisenblaetter, Clarisse Madiouni, Yasmine Laraki, Delphine Capdevielle, Stéphane Raffard

**Affiliations:** 1Faculty of Psychology, Univ Paul Valéry Montpellier 3, Univ. Montpellier, Laboratory EPSYLON EA 4556, 34090 Montpellier, France; margot.eisenblaetter@gmail.com (M.E.); madiouni.clarisse@gmail.com (C.M.); y-laraki@chu-montpellier.fr (Y.L.); 2University Department of Adult Psychiatry, CHU Montpellier, University Montpellier, Hôpital La Colombière, 34090 Montpellier, France; d-capdevielle@chu-montpellier.fr; 3Inserm, Neuropsychiatry: Epidemiological and Clinical Research, University of Montpellier, 34090 Montpellier, France

**Keywords:** vaccine, vaccination attitudes, vaccination behaviours, vaccination intentions, scale development

## Abstract

Over the past decades, vaccination has proven to be largely beneficial to global health. Despite vaccine efficacy, the French population has been recently affected by more anti-vaccination attitudes and vaccine refusal, and it is therefore necessary to validate tools to study this health issue. The Vaccination Attitudes Examination scale (VAX) is a 12-item questionnaire targeting adults that assesses general attitudes towards vaccination. The aims of the study were to translate and adapt the original English version of the scale into French and to test the psychometric properties of the scale in a French-population-based sample of adults. We included 450 French speaking adults that completed the French VAX and other questionnaires to assess convergent and divergent validities. Exploratory and confirmatory factor analyses showed that the French version of the VAX replicated the factorial structure of the original scale. Moreover, it demonstrated high internal consistency, good convergent and divergent validities, and excellent temporal stability. Furthermore, scores on the scale differentiated vaccinees from non-vaccinee respondents. Results on the scale provide us with insight into factors involved in vaccine hesitancy in France, therefore allowing French authorities and policy makers to address these specific concerns and improve vaccine acceptance rates in this country.

## 1. Introduction

Vaccination has been the one most effective measure to prevent and control many infectious diseases and has proven to be largely beneficial to global health [1]. In 1974, the World Health Organization (WHO) established a successful programme aiming to facilitate vaccination uptake for children all around the world. This programme allowed radically reducing death rates due to infectious diseases beyond borders of developing countries [2]. As a result, the WHO estimated that vaccination has saved 2–3 million children each year from life-threatening diseases. Vaccination also has substantial health-related benefits in adult populations. A study led in 2019 stated that among 42 European countries, all had established national vaccination programmes for adults [3]. For instance, influenza vaccination has proven to be beneficial for healthy working adults due to reducing frequency of respiratory illnesses, absenteeism from work, and medical visits [4].

Despite vaccine efficacy, the coverage of many recommended vaccines is still inadequate. Two types of barriers to vaccination have been identified: structural and attitudinal barriers [5]. Structural barriers correspond to “systemic issues that may limit the ability of individual persons to access a vaccine service”, whereas attitudinal barriers refer to vaccine hesitancy or “delay in acceptance or refusal of vaccines despite the availability of vaccination services” [6].

Regarding vaccination confidence, a large-scale study in 2015 measuring worldwide variations in attitudes about vaccination showed that the European region reported the highest mean-averaged negative responses for vaccine importance, safety, and effectiveness [7]. Despite the widespread access to vaccines, France was the least confident country: 41% of French people considered vaccines unsafe, representing a much higher rate than the global average (13%). Controversies related to the various vaccination campaigns in France since the 2000s may justify the presence of anti-vaccination attitudes in this country [8]. For instance, a study showed that the 2009 influenza A(H1N1) crisis contributed to a general increase in negative attitudes towards vaccination in France [9].

Numerous studies showed that attitudes towards vaccination and actual vaccination behaviours are strongly associated [9,10,11,12]. Hence, the loss of confidence in vaccine importance, safety, and effectiveness over the past few years has led to a decline in vaccination coverage against certain diseases for which vaccination is recommended [13], leading to vaccine-preventable disease outbreaks [14,15]. For example, the measles–mumps–rubella vaccination coverage did not reach the optimal level, thus causing a measle outbreak in France in 2008 [16]. In the current context of the COVID-19 pandemic, vaccination has been one of the major disease control strategies. However, France had one of the lowest COVID-19 vaccine acceptance rates worldwide, and this could be linked to a lack of confidence in the safety of vaccines [17].

The WHO listed vaccine hesitancy as the eighth major threat for global health in 2019 [18], and their Strategic Advisory Group of Experts has called for better monitoring of vaccine hesitancy [19]. For this purpose, it is necessary to develop and validate tools. Providing insight into factors involved in vaccine hesitancy could enable health authorities to address the issue with responses to specific concerns of the studied populations. Existing scales are mostly aimed at parents, as a large proportion of vaccinations are given during childhood. For example, the Parent Attitudes about Childhood Vaccines survey [20] assesses parental vaccine hesitancy. Other scales focus on specific vaccines, such as the Measles Vaccine Hesitancy scale [21].

The Vaccination Attitudes Examination (VAX) scale is a 12-item questionnaire that assesses general attitudes towards vaccination in adults [22]. It contains four subscales that enable a thorough understanding of the nature of those views: mistrust of vaccine benefits, worries about unforeseen future effects, concerns about commercial profiteering, and preference for natural immunity. A strength of the VAX scale is that its scores have been shown to be significantly associated with previous vaccination behaviours and intentions to receive future vaccines [22,23,24]. Higher scores on all subscales were found to be associated with higher risks of being unwilling to receive a COVID-19 vaccine in 2020 [25]. Therefore, the VAX scale is a powerful tool to differentiate vaccinees from respondents who are more likely to refuse vaccination.

The VAX scale has been validated in several languages including Turkish [26], Spanish [23], and Romanian [27]. There is currently no French version of the VAX scale, although French is the fifth most widely spoken language in the world. Moreover, the low COVID-19 vaccine acceptance rate found in France has highlighted the necessity to have a reliable scale to evaluate attitudes towards vaccination in a French population, especially when vaccination attitudes are strong predictors of vaccination behaviours [17,25].

Consequently, the aims of this study were to translate and adapt the VAX scale into French and to test its psychometric properties in a French-population-based sample. In line with scale validation recommendations, we applied exploratory factor analysis followed by confirmatory factor analysis to validate the factorial structure of our French VAX scale [28,29]. Finally, we assessed internal consistency, temporal stability, and convergent and divergent validities. We predicted that the French VAX scale would replicate the factorial structure of the original scale and would have good psychometric qualities.

## 2. Materials and Methods

### 2.1. Participants

In total, 612 participants were registered on our database, including both online and face to face participants. Part of the recruitment was carried out online using ads on social media platforms (Facebook, Linkedin, Instagram, Twitter). For the rest, acquaintances were asked to spread the word. Inclusion criteria were speaking French fluently and age 18 years and over.

### 2.2. Procedure

#### The VAX Scale Translation Process

The VAX scale was translated and adapted into French following the backward and forward method [30]. The authors gave us authorization to conduct the French validation of the tool. Then, two bilingual translators translated the tool from English into French. Another expert resolved any discrepancies between the versions into a single translation. Finally, a bilingual translator, blinded to the initial survey items, made a back-translation, and a comparison with the original version was conducted to verify that both versions were equivalent. In the end, a harmonized version of the tool in the target language was proposed.

Regarding response modality, we converted the initial 6-point Likert-type scale into a 5-point Likert-type scale, ranging from “Strongly disagree” to “Strongly agree”. Indeed, scientific literature shows that a scale with an odd number of response possibilities allows for the middle point to be used as a neutral point and omitting the midpoint may reduce validity when respondents lack knowledge on which to base their response [31,32].

### 2.3. Experimental Design

The purpose of the study was explained to participants, and full written consent was obtained. Participants could then access the different questionnaires. These steps could be carried out either online (using Qualtrics software) or face-to-face (using paper questionnaires). In order to measure temporal validity, we invited participants to complete the VAX a second time, one month after the first completion.

Institution’s ethics committee approval was granted for the study (IRB Accreditation number: 202100938).

### 2.4. Questionnaires

Participants were asked for basic demographic and clinical information: sex, age, education level, occupational status, whether participants have children or not, the presence of any chronic diseases, and medical treatment.

#### 2.4.1. French Version of the Vaccination Attitudes Examination (VAX) Scale

The VAX scale was originally developed in English [22]. This scale contains 12 statement-like items measuring general attitudes towards vaccination (e.g., “I feel safe after being vaccinated”). In our adapted version, participants were asked to respond on a 5-point Likert-type scale ranging from “Strongly disagree” to “Strongly agree”. Items 1, 2, and 3 are reverse-coded. Total scores range from 12 to 60, higher scores reflecting stronger anti-vaccination attitudes. In addition, four subscores may be calculated by adding the following items: (a) 1, 2, and 3 for “Mistrust of vaccine benefit”; (b) 4, 5, and 6 for “Worries about unforeseen future effects”; (c) 7, 8, and 9 for “Concerns about commercial profiteering”; (d) 10, 11, and 12 for “Preference for natural immunity”. Scores on each subscale range from 3 to 15, whereby higher scores reflect (a) more mistrust, (b) more worries, (c) more concerns, and (d) higher preference for natural immunity.

#### 2.4.2. Vaccination Behaviours and Intentions

Vaccination behaviours were assessed with two items asking whether respondents had received an influenza shot and a COVID-19 shot during the previous year. Response format was dichotomous “Yes/No”. Vaccination intentions were assessed with two items asking whether respondents had planned to receive an influenza or COVID-19 shot in the coming year. Response format was “Yes/No/Maybe/Already done”.

#### 2.4.3. Vaccine Conspiracy Beliefs Scale

The Vaccine Conspiracy Beliefs Scale (VCBS) was developed in English and French [33], to assess the presence of vaccine-specific conspiracy beliefs. It contains 7 items (e.g., “Vaccine safety data is often fabricated”) for which participants are asked to respond on a 7-point Likert scale ranging from “Strongly disagree” to “Strongly agree”. An average score is calculated, higher scores reflecting stronger conspiracy beliefs.

#### 2.4.4. Beliefs about Medicines Questionnaire

The Beliefs about Medicines Questionnaire (BMQ) assesses cognitive representations of medication and was validated in French [34]. Participants are asked to respond to 18 items on a 5-point Likert scale ranging from “Strongly agree” to “Strongly disagree”. It comprises two sections. The BMQ—Specific section contains 10 items, which measure (a) the perceived necessity for a prescribed treatment (5 items, e.g., “My health, at present, depends on my medicines”) and (b) potential concerns about its negative effects (5 items, e.g., “My medicines disrupt my life”). By adding the reverse scores of each of the 5 items, this section allows us to calculate 2 scores between 5 and 25. The BMQ—General section comprises 8 items measuring (a′) responders’ beliefs about the way doctors prescribe medicines (4 items, e.g., “Doctors place too much trust on medicines”) and (b′) the potential negative effects of treatment in general (4 items, e.g., “Medicines do more harm than good”). By adding the reverse scores of each of the 4 items, this section allows us to calculate 2 scores between 4 and 20. Higher scores mean stronger beliefs. The two sections can be administered separately, so the BMQ—Specific subscale was administered to participants who previously declared that they had a chronic disease for which they needed to take a specific treatment. The BMQ—General subscale was administered to every participant.

#### 2.4.5. Parent Attitudes about Childhood Vaccines Survey

The Parent Attitudes about Childhood Vaccines survey (PACV) assesses parental vaccine hesitancy and was validated in French in 2021 [35]. It contains 15 items with different response formats (dichotomous, “Yes/No”; 5-point Likert scale, e.g., ranging from “Strongly agree” to “Strongly disagree”; and 10-point Likert scale, e.g., ranging from “Not sure at all” to “Completely sure”). According to the method used by the initial authors [20], these formats were then collapsed into 3 response categories: hesitant (receiving a score of 2), not sure/don’t know (receiving a score of 1), and not hesitant (receiving a score of zero). By adding the score of the items, we obtained a total score ranging from 0 to 30.

#### 2.4.6. Survey Validity Check

Conducting surveys online exposes researchers to more responder carelessness and fraud [36]. Even low levels of these can lead to a decrease in the quality of collected data. As recommended, six validated attention-check items were added across the online surveys to detect careless or fraudulent responding. Four items were added to the first form and two to the second form (i.e., the form whose purpose is to test the temporal reliability of the tool), such as “Please answer “Strongly disagree” to this question” [37]. In this case, any other response than “Strongly disagree” is considered incorrect. Correct answers received a score of zero and incorrect responses a score of one. A total score out of 4 was computed for the first form and a total score out of 2 for the second form.

## 3. Results

### 3.1. Statistical Analyses

Statistical analyses were carried out using IBM SPSS software version 24.0, New York, USA. We decided to remove participants from the study based on the following criteria: not having responded to the totality of the validity-check items, having responses that did not pass the validity check, having completed the study twice on the online survey, and having abandoned the completion of the form before the end. The winzoring method was used to process outlier scores for our variables of interest [38]. No normal distribution was considered when absolute values for skewness and kurtosis were greater than 3 and 10, respectively [38]. Means and standard deviations were computed for continuous variables, and categorical variables were expressed in percentages.

An exploratory factor analysis (EFA) was conducted on the 12 items of the scale. The Kaiser–Meyer–Olkin (KMO) test was used to verify the sampling adequacy. A KMO > 0.50 indicates an adequate sample to run an EFA [38]. To test intercorrelation between VAX scale items, Bartlett’s test of sphericity was computed, for which a *p*-value below the 0.05 threshold indicates a good correlation between the variables. As our data were normally distributed, the maximum likelihood extraction method was chosen [39]. The choice of the number of factors to be extracted was based on parallel analysis and scree plot [40,41]. The parallel analysis was performed by randomly generating a dataset with the same number of observations and variables as in the original data. The recommended number of factors to extract is the number of original eigenvalues that are greater than their respective 99th percentile of simulated values [29,39]. The number of factors was determined by locating the point where the slope of the scree plot curve stabilises below the inflexion point. As our factors are expected to be correlated with each other, the oblimin rotation was selected. After rotation, items with loadings >0.32 were considered statistically meaningful factors. Items with loadings <0.32 and cross loadings >0.32 were considered as suspect.

To test the factorial structure of the VAX scale obtained from the EFA, a confirmatory factor analysis (CFA) was performed using IBM SPSS Amos 26.0, New York, NY, USA. The model fit was assessed with the ratio of c2 to degrees of freedom (χ2/df, *p* > 0.05), the Tucker Lewis index (TLI > 0.90), the comparative fit index (CFI > 0.90), and the root mean square error of approximation (RMSEA < 0.08) [42]. The model selection was made based on the Expected Cross-Validation Index (ECVI). The lower the ECVI value, the better the performance of the model [43].

The internal consistency of the VAX scale was assessed with McDonald’s Omega coefficient. Any values below 0.70 reflect a lack of reliability of the tool [44].

Convergent validity was examined with Pearson correlation between the VAX total score and scores of the PACV survey, the BMQ, and the VCBS. A correlation coefficient between 0.10 and 0.30 indicates a small effect size, a correlation coefficient between 0.30 and 0.50 indicates a medium effect, and a correlation coefficient above 0.50 indicates a large effect [45]. T-tests for independent samples were conducted to compare the VAX total score between (a) participants who received an influenza vaccine during the past year and those who did not, and (b) participants who received a COVID-19 vaccine during the past year and those who did not. Analyses of variance were carried out to compare the VAX total score between (c) participants who intended to receive an influenza shot in the coming year, those who hesitated (i.e., “Maybe” response), and those who did not; and (d) participants who intended to receive a COVID-19 shot in the coming year, those who hesitated, and those who did not. Given that sample sizes were unequal for each analysis, we adjusted sample size by randomly selecting a number of participants approximately equal to that of the smallest sample. If the analyses revealed any significant differences between groups, post hoc analyses were performed with the Bonferroni correction. Finally, the divergent validity was conducted using correlational analyses between the VAX total score and age and years of scholarship. Gender effect was explored with Student’s t-test for independent samples.

To establish the temporal stability of the VAX scale, the intraclass correlation coefficient (ICC) was computed. An ICC below 0.50 reflects poor reliability, an ICC between 0.5 and 0.75 reflects moderate reliability, an ICC between 0.75 and 0.90 reflects good reliability, and an ICC above 0.90 excellent reliability [46].

All analyses were conducted with a significance threshold of α ≤ 0.05, two-tailed.

### 3.2. Results

#### 3.2.1. Preliminary Analysis

From our initial sample (*n* = 612), we removed 162 participants for the following reasons: 132 participants did not respond to the totality of the validity-check items, 3 respondents completed the study twice on the online survey, 3 respondents abandoned the completion of the form before the end, and 24 respondents did not pass the validity check.

Regarding the latter, those who received a score > 1 were excluded, representing 9% of the total sample (*n* = 24). In the final sample (*n* = 450), only 3 values were identified as outlier data, representing 0.04% of total variables. All data from questionnaires had satisfactory skewness and kurtosis values, suggesting a normal distribution.

The total sample was randomly split in two groups. The sub-sample 1 (*n* = 225) was used to conduct the EFA, and the sub-sample 2 (*n* = 225) was used to conduct the CFA. Sub-samples presented no significant differences in terms of age (t447= 0.81, *p* = 0.42), sex proportion (χ2 = 0.29, *p* = 0.59), years of education (t447 = 0.19, *p* = 0.85), or experimental modality (Qualtrics vs. paper, χ2 = 0.04, *p* = 0.83).

#### 3.2.2. Demographic and Clinical Characteristics

Descriptive statistics for all demographic and clinical characteristics for the total sample and the two sub-samples are presented in Table 1. Our total sample had a mean age of 32.92 years (±14.16), and females represented 75% of our total sample group. Given that vaccine refusal and vaccine hesitancy have both been shown to be significantly associated with female sex in the literature [47,48], we decided to perform a weighted adjustment based on participants’ sex. To do so, the proportion of females was adjusted from 75% to 50%, and the proportion of males was adjusted from 25% to 50%. Weighting coefficients were 0.66 for females and 2 for males and were assigned to each variable of our dataset. All statistical analyses were then performed on the weighted data.

#### 3.2.3. Exploratory Factor Analysis

The EFA was conducted on sub-sample 1. Descriptive statistics of the VAX scale items and results of the EFA are documented in Table 2. Correlation matrix suggested no problem of multicollinearity. All items presented correlation coefficients >0.30 with the other items, except for item 4, which demonstrated lower but nonetheless sufficient correlation coefficients with the other items (i.e., r > 0.16). However, we decided to keep item 4, as it may be clinically relevant. The overall KMO value was meritorious (0.89), and Bartlett’s test of sphericity was significant (χ2 = 1899, df = 66, *p* < 0.001), suggesting that the sample was appropriate to run an EFA. The parallel analysis indicated four factors with eigenvalues higher than the stimulated data (factor 1, 6.45 versus 0.61; factor 2, 0.79 versus 0.46; factor 3, 0.57 versus 0.36; factor 4, 0.46 versus 0.27). The next six factors were not supported (factor 5, 0.01 versus 0.19; factor 6, −0.00 versus 0.12; factor 7, −0.02 versus 0.06; factor 8, −0.07 versus 0.01; factor 9, −0.10 versus −0.05; factor 10, −0.12 versus −0.11) (Figure 1). The scree plot indicated a point of inflexion at five factors. The parallel analysis and the scree plot both suggested a 4-factor structure, explaining 80.23% of the total variance. Based on these results, a forced 4-factors EFA was performed on the VAX. As found in Table 2, item 6, “I worry about the unknown effects of vaccines in the future”, loaded on factors 4 and 1. According to the standard recommendations, this item should be removed from the scale. However, when we removed item 6, items 4 and 5 were no longer significant. These three items referred to the dimension “Worries about unforeseen future effects”. Still, a study led during the COVID-19 pandemic showed that concerns about unknown long-term effects of vaccines were the primary reason for vaccine hesitancy [49], reflecting the importance of retaining these items. Another study found similar results: scores of the “Mistrust of vaccine benefits” and “Worries about unforeseen future effects” VAX scale dimensions were found to be the most important determinants of uncertainty and unwillingness to vaccinate against COVID-19 [25]. Because item 6 directly concerns the harmful long-term effects of vaccines, we decided to keep it in the factor containing items 4 and 5, which explore similar beliefs (i.e., unknown problems with vaccines and potential problems for children, respectively). To conclude, we found the “Mistrust of vaccine benefits” factor, composed of items 1, 2, and 3; the “Worries about unforeseen future effects” factor, composed of items 4, 5, and 6; the “Concerns about commercial profiteering” factor, composed of items 7, 8, and 9; and the “Preference for natural immunity” factor, composed of items 10, 11, and 12. All factors are significantly correlated with each other (factor 1 and 2, r = −0.67; factor 1 and 3, r = 0.67; factor 1 and 4, r = 0.39; factor 2 and 3, r = −0.61; factor 2 and 4, r = −0.34; factor 3 and 4, r = 0.43).

#### 3.2.4. Confirmatory Factor Analysis

The CFA was conducted on sub-sample 2. Two models were computed: the 4-factor model previously established with EFA, and a single-factor model. In the 1-factor model, all VAX scale items were an indicator of one unique latent variable. The 4-factor model showed a good fit with regards to model fit index (χ^2^/df = 2,65, *p* < 0.001; TLI = 0.94; CFI = 0.96; RMSEA = 0.08), while the 1-factor model showed a less satisfactory adjustment (c2/df = 10.06, *p* < 0.001; TLI = 0.69; CFI = 0.75; RMSEA = 0.20). Moreover, comparison of the two models, based on the ECVI, showed that the 4-factor model is a better fit (4-factor model ECVI = 0.91; 1-factor model ECVI = 2.85). Figure 2 illustrates the path diagram for the retained model.

#### 3.2.5. Internal Consistency

The internal consistency was assessed using the total sample. The results reveal high internal consistency of the VAX total score (w = 0.93) and the four factors (“Mistrust of vaccine benefit”, w = 0.91; “Worries about unforeseen future effects”, w = 0.79; “Concerns about commercial profiteering”, w = 0.90; “Preference for natural immunity”, w = 0.87).

#### 3.2.6. Convergent Validity

Means and standard deviations of questionnaires are reported in Table 3. Descriptive characteristics of the VAX total score by gender and age categories are documented in Appendix A.

Correlational analyses revealed that all VAX scores were strongly associated with the VCBS total score and the PACV survey total score. Moderate to strong correlations were found between all VAX scores, the BMQ—General scores, and BMQ—Concerns about its negative effects. BMQ—Perceived necessity for a prescribed treatment was moderately correlated with total VAX score and the “Concerns about commercial profiteering” factor. Weak correlations were observed between the “Worries about unforeseen future effects” factor, the “Preference for natural immunity” factor, and BMQ—Perceived necessity for a prescribed treatment. Finally, no correlation was found between the “Mistrust of vaccine benefit” factor and BMQ—Perceived necessity for a prescribed treatment (Table 4).

The participants who had not received influenza and COVID-19 vaccines in the previous year had a higher VAX total score (respectively, 27.90 ± 9.15; 40.43 ± 12.38) compared to those who were vaccinated (respectively, 24.34 ± 9.09; 27.82 ± 8.64) (respectively, t102 = 1.98, *p* = 0.05, d′ = 0.19; t197 = 8.31, *p* < 0.001, d′ = 0.51). Regarding influenza vaccination intentions, a significant group effect on the VAX total score was observed (F = 14.93, *p* < 0.001, *n^2^* = 0.23). Post hoc analyses indicated that the participants that did not plan on receiving an influenza shot in the coming year had a higher VAX total score (35.57 ± 13.75) compared to those who hesitated (25.60 ± 7.22) and compared to those who did plan to be vaccinated (23.24 ± 7.16) (*p* < 0.001). There was no significant difference between participants who planned on receiving an influenza shot and those who hesitated (*p* = 0.94). Regarding COVID-19 vaccination intentions, a significant group effect was observed on the VAX total score (F = 62.62, *p* < 0.001, n2 = 0.45). Post hoc analyses indicated that participants that did not plan on receiving a COVID-19 shot in the coming year had a higher VAX total score (44.83 ± 9.33) compared to those who hesitated (31.92 ± 8.66) and did plan to be vaccinated (27.18 ± 7.68) (*p* < 0.001). Participants who hesitated to be vaccinated for the COVID-19 had a higher VAX total score compared to those who did plan to be vaccinated (*p* < 0.02).

#### 3.2.7. Divergent Validity

Divergent validity was assessed using the total sample. Results show no significant correlation between age and the VAX total score (r = 0.05). A weak correlation was found between years of education and the VAX total score (r = −0.29, *p* < 0.001). Women reported a higher VAX total score than men with a small effect size (women, 32.63 ± 11.34, versus men, 28.65 ± 10.29, t448 = −3.98, *p* < 0.001, d’ = 0.18).

#### 3.2.8. Temporal Stability

A total of 168 participants completed the VAX scale twice. Descriptive statistics for both administrations are presented in Table 5. The ICC computed between the VAX total scores was significant (r = 0.95 with a 95% confidence interval from 0.94 to 0.97; *p* < 0.001), reflecting excellent reliability. A paired t-test revealed that the VAX total score did not differ significantly from the first administration (28.45 ± 10.36) to the second (28.54 ± 10.23) (t169 = −0.28, *p* = 0.78).

## 4. Discussion

The goals of the present study were to translate and adapt the original English version of the Vaccination Attitude Examination (VAX) scale into French and to test the psychometric properties of this scale in a French-population-based sample.

The exploratory factor analysis showed a 4-factor solution with 12 items that explained 80.23% of the variance. The four factors are “Mistrust of vaccine benefits”, “Worries about unforeseen future effects”, “Concerns about commercial profiteering”, and “Preference for natural immunity”. Moreover, the French adaptation of the VAX scale demonstrated high internal consistency reliability and a strong temporal stability.

Further statistical analyses provided more arguments to establish validity, in particular convergent validity. The literature shows that endorsing vaccine conspiracy beliefs predicts general attitudes towards vaccines [50]. This relationship is verified in our study, as our analyses showed a positive correlation between total scores of the VAX scale and those of the Vaccine Conspiracy Beliefs Scale. We also found positive correlations between total scores of the VAX scale and all those of the Beliefs about Medicine Questionnaire (BMQ) subscales. These findings are in line with previous conclusions of a replication study assessing validity of the VAX scale and the Spanish validation of the scale [23,24]. We also tested the relationship between the total scores of the VAX scale and those of the Parent Attitudes about Childhood Vaccines (PACV) survey. As expected, we found a positive correlation between the two scores. This correlation replicates results found in the original validation paper [22].

Attitudes towards vaccination and actual vaccination behaviours have been found to be strongly associated in numerous studies [9,10,11,12]. This is reflected in our results, as we demonstrated that the total score of the VAX scale successfully differentiated both COVID-19 and influenza vaccinees from non-vaccinees. Indeed, participants who did not receive these vaccines in the previous year manifested stronger anti-vaccination attitudes than those who did. A similar pattern was found for vaccination intentions: participants who did not plan on being vaccinated against influenza in the coming year manifested stronger anti-vaccination attitude than those who did and those who hesitated. The relationship between attitudes and vaccination intentions is even stronger regarding the COVID-19 vaccine because the VAX scale is able to differentiate all three groups of respondents: those who planned on being vaccinated, those who did not, and those who hesitated. These results highlight that the VAX scale is a powerful tool to differentiate vaccinees from respondents who are more likely to refuse to be vaccinated. Moreover, our results replicate the findings of the other VAX scale validation papers [22,23,24].

In terms of divergent validity, total scores of the VAX scale are unrelated to age of the participants. Regarding sex effect, it appears that females reported stronger antivaccination attitudes than males. This is in line with recent findings in the literature [17,47,48]. Even though the VAX scale scores were unrelated to level of schooling in the original validation paper [22], a small correlation was also found between educational level and total scores of the VAX scale. However, the authors mentioned that their results may be related to the restricted range of education of their samples. Our result is not surprising. as such findings have been found in other studies: more vaccine hesitancy seems to be associated with lower educational levels [47,48]. Moreover, in the replication validation study conducted by Wood et al. [24], the relation between VAX score and education level is also found: increased education predicted lower VAX scores.

Nevertheless, the results of this study should be considered carefully. Conducting online surveys has many advantages, including automation, speed, and reach [51]. In this study, online recruitment allowed us to quickly recruit many participants via various social networks. In addition, some of the data entry work is automated, thus avoiding researchers committing data entry errors. However, when conducting an Internet-based survey, researchers should be aware of the pitfalls of this type of research. The most important disadvantages are selection biases: under-coverage and self-selection [52,53]. With regard to under-coverage, it should be noted that the Internet has become accessible to a wide range of different people [54], and we made sure to publish the advertisement on various social networks (Facebook, LinkedIn, Instagram, Twitter) to reach users of different types. Self-selection bias can lead to a non-representative sample, as individuals who choose to participate may be more motivated, have stronger opinions or experiences, or possess different characteristics than those who do not participate [52]. For instance, women tend to be more likely to select themselves into research projects [55], and that can explain why our sample is mostly composed of women. We minimised this particular imbalance by making a weighted adjustment of our data based on participants’ sex. Moreover, to address both under-coverage and self-selection, part of the recruitment was conducted using paper-and-pencil surveys in addition to Internet-based recruitment. In this way, people without Internet or social media were not excluded from the study. In addition, this allowed for a more random selection of participants and minimized self-selection bias, thus ensuring a more representative sample of the French population. Ultimately, conducting surveys online exposes researchers to more responder carelessness and fraud [36]. Even though attention-check items were added across the online surveys to control for careless or fraudulent responding, these measures carry risks. For instance, attention checks may influence responders’ further responses because of their obvious character that informs responders about researchers’ intention to evaluate their attention level and put them in a more deliberative mindset [56]. Nevertheless, the incorporation of instructed response items has proven to ensure high-quality data in previous research [37,57]. Furthermore, Kung et al. [56] specifically investigated whether validity items influence participants’ responses, and they found no evidence that they affect scale validity. Additionally, we can assume that the validity checks did not compromise the validity of the scale but rather helped us make appropriate data quality decisions.

The current pandemic context introduced another bias: although our scale is supposed to assess general attitudes towards vaccination, vaccination is currently a worldwide hot topic. Thus, there is a risk that participants responded to our survey in reference to the COVID-19 vaccine only. Even though the emphasis was placed on this distinction during the face-to-face interviews, this was not possible for online participants. Another limit of this study is that we did not test whether this factor structure held for clinical populations.

Finally, given the high proportion of women, we decided to make a weight adjustment based on participant sex. Thus, we followed the Classical Test Theory (CTT) and not consider the Item Response Theory (IRT). Nevertheless, empirical studies comparing the CTT and other statistical frameworks such as the item response theory (IRT) suggest that the CTT may be as effective as IRT for assessing individual change, especially when the scale contains fewer than 20 items, which is the case with the VAX scale [58,59,60].

## 5. Conclusions

To conclude, the French version of the VAX scale is a simple tool that demonstrated strong results in the various statistical analyses carried out to validate the tool. The validation of the French VAX scale enables researchers to identify groups of particular concern with respect to vaccine hesitancy and to determine specifically which aspect of vaccination is especially problematic among these groups. The French population is particularly affected by anti-vaccination attitudes, and this tool allows studying the dynamics involved in this health issue. Further understanding of attitudes towards vaccination will enable healthcare institutions to directly address populations’ specific concerns about vaccination and, therefore, improve vaccine acceptance rates as a beneficial consequence.

## Figures and Tables

**Figure 1 vaccines-11-01001-f001:**
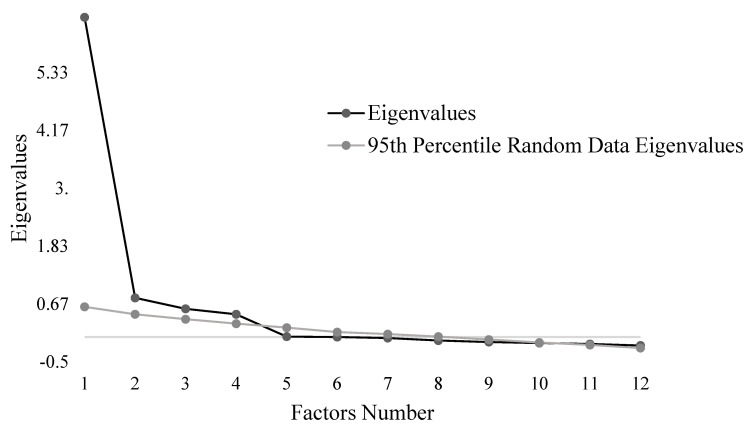
Screen plot and parallel analysis of eigenvalues for the Vaccination Attitudes Examination factors.

**Figure 2 vaccines-11-01001-f002:**
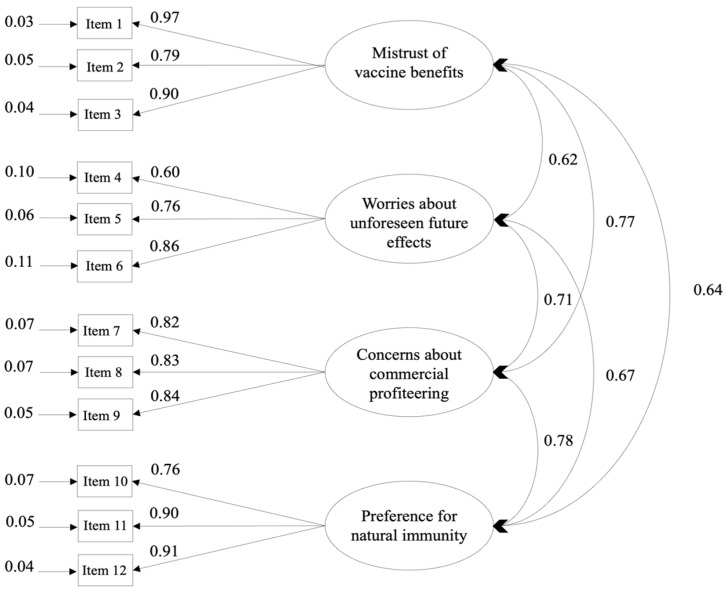
Path diagram for the Vaccination Attitudes Examination factors.

**Table 1 vaccines-11-01001-t001:** Demographic characteristics of participants.

	Total Sample (*n* = 450)	Sub-Sample 1 (*n* = 225)	Sub-Sample 2 (*n* = 225)
*Experimental modality*	-	-	-
Qualtrics	72%	71%	72%
Paper	28%	29%	28%
*Demographics*	-	-	-
Age, years	32.92 ± 14.16 [18–81]	33.46 ± 14.64 [18–80]	32.38 ± 13.66 [18–81]
Female	75%	74%	76%
Years of scholarship	15.05 ± 2.28 [9–22]	15.07 ± 2.28 [9–20]	15.04 ± 2.29 [9–22]
*Occupational status*	-	-	-
Working	52%	53%	51%
Student	35%	33%	38%
Retired	6%	7%	4%
Unemployed	6%	6%	6%
Disabled worker or unemployed with disabled status	1%	1%	1%
*Family life*	-	-	-
Have at least one child	29%	32%	27%
*Health*	-	-	-
Have a chronic disease	20%	21%	18%
Take a treatment	20%	23%	18%
*Past vaccination behaviours*	-	-	-
Influenza vaccination	-	-	-
Yes	11%	14%	8%
No	89%	86%	92%
COVID-19 vaccination	-	-	-
Yes	77%	76%	78%
No	23%	24%	22%
*Vaccination intentions*	-	-	-
Influenza vaccination	-	-	-
Yes	8%	9%	6%
No	72%	70%	74%
Maybe	16%	16%	16%
Already done	4%	5%	4%
COVID-19 vaccination	-	-	-
Yes	28%	29%	28%
No	17%	17%	17%
Maybe	12%	13%	10%
Already done	43%	41%	45%

Note. Data are presented as means ± standard deviations [range], *n* = number of participants.

**Table 2 vaccines-11-01001-t002:** Mean, standard deviation, and factor loadings for the Vaccination Attitudes Examination scale.

Items	Mean ± SD	Factor 1	Factor 2	Factor 3	Factor 4
1	2.50 ± 1.19	0.03	**−0.90**	−0.03	0.06
2	1.85 ± 1.01	0.19	**−0.52**	0.18	−0.12
3	2.35 ± 1.17	−0.07	**−1.00**	−0.02	0.02
4	3.65 ± 1.14	−0.03	−0.01	−0.01	**0.86**
5	3.14 ± 1.18	0.19	−0.05	0.26	**0.46**
6	2.70 ± 1.37	**0.35**	−0.18	0.11	**0.32**
7	2.29 ± 1.29	**0.79**	−0.01	0.04	0.02
8	2.23 ± 1.36	**0.95**	−0.002	−0.11	0.04
9	1.79 ± 1.22	**0.79**	−0.04	0.15	−0.08
10	2.88 ± 1.21	−0.002	−0.02	**0.73**	−0.01
11	2.59 ± 1.34	−0.03	−0.05	**0.86**	0.01
12	2.31 ± 1.26	0.04	−0.04	**0.83**	0.05
% of variance		55.41	9.73	8.09	7

Note. Data are presented as means ± standard deviations [range]. Factor loading of at least 0.32 appears in bold.

**Table 3 vaccines-11-01001-t003:** Mean and standard deviation of questionnaires.

	Mean ± SD [Min–Max]
*Vaccination Attitudes Examination Scale* ^α^	-
Total	30.68 ± 11.01 [12–60]
Mistrust of vaccine benefits	6.89 ± 3.22 [3–15]
Worries about unforeseen future effects	9.64 ± 3.11 [3–15]
Concerns about commercial profiteering	6.42 ± 3.47 [3–15]
Preference for natural immunity	7.72 ± 3.39 [3–15]
*Vaccine Conspiracy Belief Scale* ^α^	2.88 ± 1.40 [1–7]
*Parent Attitudes about Childhood Vaccines Survey* ^β^	9.24 ± 7.93 [0–26]
*Beliefs about Medicines Questionnaire*	-
Specific ^δ^	-
Perceived necessity for a prescribed treatment	10.7 ± 3.93 [5–25]
Concerns about negative effects	17.9 ± 4.19 [8–25]
General ^α^	-
Beliefs about prescribe medicines	12 ± 3.48 [4–20]
Perceived negative effects of treatment	15.13 ± 3.23 [4–20]

Note. SD = standard deviation, *n* = number of participants. As the Parent Attitudes about Childhood Vaccines Survey and the Belief about Medicines Questionnaires—Specific did not cover all participants, the number of participants for these questionnaires is different from the total sample. We reported the weighted number of participants. ^α^ *n* = 447, ^β^ *n* = 129, ^δ^ *n* = 70.

**Table 4 vaccines-11-01001-t004:** Correlational analyses between questionnaires.

	*Vaccination Attitudes Examination Scale* ^α^				
	Total	Mistrust of Vaccine Benefit	Worries about Unforeseen Future Effects	Concerns about Commercial Profiteering	Preference for Natural Immunity
*Vaccine Conspiracy Belief Scale* ^α^	0.84 **	0.71 **	0.63 **	0.78 **	−0.68 **
*Parent Attitudes about Childhood Vaccines Survey* ^β^	0.84 **	0.66 **	0.73 **	0.78 **	−0.73 **
*Beliefs about Medicines Questionnaire*	-	-	-	-	-
Specific ^δ^	-	-	-	-	-
Perceived necessity for a prescribed treatment	0.32 **	−0.23	−0.27 *	−0.33 **	−0.28 *
Concerns about its negative effects	−0.46 **	−0.35 *	−0.40 **	−0.40 **	−0.42 **
General ^α^	-	-	-	-	-
Beliefs about prescribe medicines	0.58 **	−0.47 **	−0.43 **	−0.54 **	−0.49 **
Perceived negative effects of treatment	−0.54 **	−0.41 **	−0.39 **	−0.53 *	−0.46 **

Note. *p* < 0.05 *, *p* < 0.001 **. As the Parent Attitudes about Childhood Vaccines Survey and the Belief about Medicines Questionnaires—Specific did not cover all participants, the number of participants for these questionnaires is different from the total sample. We reported the weighted number of participants. ^α^ *n* = 447, ^β^ *n* = 129, ^δ^ *n* = 70.

**Table 5 vaccines-11-01001-t005:** Mean and standard deviation of the 1st and 2nd administration of the Vaccination Attitudes Examination scale.

	Mean ± Standard Deviation	
Items	1st Administration Mean ± SD	2nd Administration Mean ± SD
1	2.40 ± 1.15	2.34 ± 1.1
2	1.83 ± 1.01	1.79 ± 0.91
3	2.31 ± 1.09	2.29 ± 1.02
4	3.47 ± 1.19	3.62 ± 1.13
5	3.07 ± 1.11	3.04 ± 1.11
6	2.53 ± 1.37	2.62 ± 1.29
7	2.24 ± 1.25	2.12 ± 1.18
8	2.01 ± 1.29	2.12 ± 1.24
9	1.59 ± 1.06	1.64 ± 1.08
10	2.62 ± 1.18	2.65 ± 1.24
11	2.31 ± 1.25	2.36 ± 1.21
12	2.05 ± 1.15	2.04 ± 1.16
Total VAX score	28.45 ± 10.36	28.54 ± 10.24
Mistrust of vaccine benefits	6.54 ± 2.98	6.41 ± 2.68
Worries about unforeseen future effects	9.09 ± 3.10	9.28 ± 2.95
Concerns about commercial profiteering	5.84 ± 3.25	5.87 ± 3.15
Preference for natural immunity	6.97 ± 3.17	6.97 ± 3.25

Note. Data are presented as means ± standard deviations [range].

## Data Availability

The anonymized dataset is available from the corresponding author on reasonable request.

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
