# Peer review of "Adaptation and Validation of a French Version of the Vaccination Attitudes Examination (VAX) Scale"

_vaccines, 2023, doi:10.3390/vaccines11051001_

Round 1

Reviewer 1 Report

Dear Editor,

Dear Authors,

Thank You for the opportunity to review a manuscript entitled Adaptation and validation of a French version of the 2 Vaccination Attitudes Examination (VAX) scale.

The manuscript is well written and in general fits the quality criteria to be published in Vaccines.

Among issues I suggest considering are:

*-provide in the supplementary data the final version of the scale including 5-point Likert scale answer options with words used across categories;

*-consider, please, providing descriptive characteristic of VAX scale values by gender and age categories (in a supplement?). This will help future readers to make some comparisons across different study groups/populations.

*-there is also an issue whether the mode of VAX scale implementation (online or face-to-face) impacted results. I suggest providing some data about that.

*-different type items used in the VAX scale surely provide different amount of the information about the latent construct of Vaccination Attitudes. In the proposed scale each item is equally weighted in the total score which in fact does not describe their real importance and the impact into the construct. Therefore the scale is constructed following the Classical Test Theory but does not consider the Item Response Theory. I suggest mentioning this issue under discussion (study limitations) or add the paragraph which IRT models and results.

Overall, it is a good paper.

Reviewer

Author Response

Reviewer 1

Thank You for the opportunity to review a manuscript entitled Adaptation and validation of a French version of the 2 Vaccination Attitudes Examination (VAX) scale.

The manuscript is well written and in general fits the quality criteria to be published in Vaccines.

Response: We sincerely thank the Reviewer 1 for the time he/she took to review our work and for his/her constructive comments.

Among issues I suggest considering are:

*-provide in the supplementary data the final version of the scale including 5-point Likert scale answer options with words used across categories;

Response: You will find the translated scale in the supplementary material. For the record, only responses 1 and 5 are named: strongly disagree (1), strongly agree (5).

*-consider, please, providing descriptive characteristic of VAX scale values by gender and age categories (in a supplement?). This will help future readers to make some comparisons across different study groups/populations.

Response: We thank the Reviewer 1 for this suggestion of analysis. Descriptive characteristic of the VAX total score by gender and age categories are documented in Supplementary Table, as follows: 

pastedGraphic.png

We also modified the Results section, as follows: 

Convergent validity

[…] Descriptive characteristic of the VAX total score by gender and age categories are documented in Supplementary Table […]. 

*-there is also an issue whether the mode of VAX scale implementation (online or face-to-face) impacted results. I suggest providing some data about that.

Response: We thank the Reviewer 1 for this constructive comment. To address this concern, we added some precisions  in the Discussion section, as follows : 

Discussion

[…]Nevertheless, the results of this study should be considered carefully. Conducting online surveys has many advantages, including automation, speed, and reach [51]. In this study, online recruitment allowed us to quickly recruit many participants via various social networks. In addition, some of the data entry work is automated, thus avoiding researchers to commit data entry errors. However, when conducting an Internet-based survey, researchers should be aware of the pitfalls of this type of research. The most important disadvantages are selection biases: under-coverage and self-selection [52].) [53]. With regard to under-coverage, it should be noted that the internet has become accessible to a wide range of different people [54], and we made sure to publish the advertisement on various social networks (Facebook, LinkedIn, Instagram, Twitter) to reach users from different types. Self-selection bias can lead to a non-representative sample, as individuals who choose to participate may be more motivated, have stronger opinions or experiences, or possess different characteristics than those who do not participate [52]. For instance, women tend to be more likely to select themselves into research projects [55] and that can explain why our sample is mostly composed of women. We minimised this particular imbalance by making a weighted adjustment of our data based on participants’ sex. Moreover, to address both under-coverage and self-selection, a part of the recruitment was made using paper-and-pencil surveys in addition to internet-based recruitment. In this way, people without internet or social media were not excluded from the study. In addition, this allowed for a more random selection of participants and minimized self-selection bias, thus ensuring a more representative sample of the French population. 

These references were added:

Ball, H. L. (2019). Conducting Online Surveys. Journal of Human Lactation, 35(3), 413417. https://doi.org/10.1177/0890334419848734

Bethlehem, J. (2010). Selection Bias in Web Surveys: Selection Bias in Web Surveys. International Statistical Review, 78(2), 161188. https://doi.org/10.1111/j.1751-5823.2010.00112.x

Suarez-Balcazar, Y., Balcazar, F. E., & Taylor-Ritzler, T. (2009). Using the Internet to conduct research with culturally diverse populations: Challenges and opportunities. Cultural Diversity and Ethnic Minority Psychology, 15(1), 96104. https://doi.org/10.1037/a0013179

Wellman, B. (2004). The Three Ages of Internet Studies: Ten, Five and Zero Years Ago. New Media & Society, 6(1), 123129. https://doi.org/10.1177/1461444804040633

Smith, G. (2008). Does Gender Influence Online Survey Participation?: A record-linkage analysis of university faculty online survey response behavior. ERIC Document Reproduction Service, ED 501717.

*-different type items used in the VAX scale surely provide different amount of the information about the latent construct of Vaccination Attitudes. In the proposed scale each item is equally weighted in the total score which in fact does not describe their real importance and the impact into the construct. Therefore the scale is constructed following the Classical Test Theory but does not consider the Item Response Theory. I suggest mentioning this issue under discussion (study limitations) or add the paragraph which IRT models and results.

Response: We thank the Reviewer 1 for this precious comment. We modified the Discussion

section, as follows: 

Discussion

[…]Finally, given the high proportion if women, we decided to make a weight adjustment based on participant’s sex. Thus, we followed the Classical Test Theory (CTT) and not consider the Item Response Theory (IRT). Nevertheless, empirical studies comparing the CTT and the IRT suggest that the CTT may be as effective as IRT for assessing individual change especially when the scale contains less 20 items, which is the case with the VAX (Jabrayilov et al., 2016; Brouwer, 2013; Sébille et al., 2010) […]

These references were added:

Sébille V., Hardouin J.-B., Le Neel T., Kubis G., Boyer F., Guillemin F., Falissard B. (2010). Methodological issues regarding power of classical test theory and IRT-based approaches for the comparison of patient-reported outcome measures—A simulation study. Medical Research Methods, 10, 1-10. [PMC free article] [PubMed] [Google Scholar]

Brouwer D., Meijer R. R., Zevalkink J. (2013). Measuring individual significant change on the Beck Depression Inventory-II through IRT-based statistics. Psychotherapy Research, 23, 489-501. [PubMed] [Google Scholar]

Jabrayilov R, Emons WHM, Sijtsma K. Comparison of Classical Test Theory and Item Response Theory in Individual Change Assessment. Appl Psychol Meas. 2016 Nov;40(8):559-572. doi: 10.1177/0146621616664046. Epub 2016 Sep 24. PMID: 29881070; PMCID: PMC5978722.

Overall, it is a good paper.

Response: We gratefully thank the Reviewer 1 for this comment. 

Reviewer 2 Report

Review: Adaptation and validation of a French version of the 2 Vaccination Attitudes Examination (VAX) scale 

Studies like these are, of course important and publishable, however, I have some queries and suggestions to be considered before publication. 

MAJOR CONCERN NUMBER 1: 

For Section 2.4.6 “Survey validity check”, the authors write “Conducting surveys online expose researchers to more responder’s carelessness and fraud [36]. Even low levels of those can lead to a decrease in the quality of collected data. As recommended, six validated attention check items were added across the online surveys to detect careless or fraudulent responding. Four items were added to the first form and two to the second form (i.e. the form whose purpose is to test the temporal reliability of the tool), such as “Please answer “Strongly disagree” to this question” [37]. In this case, any other response than “Strongly disagree” is considered incorrect. Correct answers received a score of zero and incorrect responses, a score of one. A total score out of 4 was computed for the first form and a total score out of 2 for the second form”.

When making use of Instructional Manipulation Checks (IMCs), one should acknowledge the limitations of including them. Oppenheimer et al. (2009) lists some of the disadvantages being that participants that are paying attention could feel insulted when running into an IMC and there might be a backlash by trying to foil the study and the concern that if an IMC is used to eliminate participants from the sample, then the external validity of the study could be harmed. 

MAJOR CONCERN NUMBER 2: 

At the preliminary analysis, the authors write: “From our initial sample (n = 612), we removed 132 participants who did not respond to the totality of the validity-check items, 3 respondents who completed the study twice on the online survey and 3 respondents who abandoned the completion of the form before the end. Regarding the validity-check, those who received a score > 1 were excluded, representing 9% of the total sample (n = 24).” 

This is all very unclear. The authors say 132 of 612 participants were removed. This is 21.6% 

of the respondents (almost a quarter!). Then the authors say: 

Of the 132: 

a) 3 respondents who completed the study twice on the online survey 

b) 3 respondents who abandoned the completion of the form before the end 

c) Regarding the validity-check, those who received a score > 1 were excluded, 

representing 9% of the total sample (n = 24)

But 3 + 3 + 24 = 30 (not 132). And there are problems with the choice of exclusion: 

For (a), how was it determined they completed the study twice? Using the computer’s IP address. This then would be wrong to do, as in a household, say the wife completed the questionnaire, then was so fascinated by / interested in it, she called her husband to complete the same questionnaire on the same computer. 

For (b), what if they completed 95% of the questionnaire and then left? I would then still advocate for using this data, as SPSS has ways of dealing with missing values (especially if the % missing values are as small as 5%). 

For (c), see my major concern number 1 of there being disadvantages to using IMCs that are not considered in the manuscript. 

MAJOR CONCERN NUMBER 3: 

The authors write, “No normal distribution was considered when absolute values for skewness and kurtosis were greater than 3 and 10, respectively [38]”. One should use proper inferential statistical tests (with p-values) to test normality, such as the Kolmogorov-Smirnov test or the Shapiro-Wilk test (p > 0.05 indicated normality). One should never use skewness and kurtosis, as there are different recommendations in the literature about the cut-off scores, and there has no been no agreement on “final” cut-off scores, i.e. one researcher might say the kurtosis cutoff is 10; someone else will say something else (there is no consensus on this in the literature). 

References 

Oppenheimer, D. M., Meyvis, T., & Davidenko, N. (2009). Instructional manipulation checks: Detecting satisficing to increase statistical power. Journal of Experimental Social Psychology, 45(4), 867-872. https://doi.org/10.1016/j.jesp.2009.03.009

/

Author Response

Reviewer 2 

Review: Adaptation and validation of a French version of the 2 Vaccination Attitudes Examination (VAX) scale 

Studies like these are, of course important and publishable, however, I have some queries and suggestions to be considered before publication. 

MAJOR CONCERN NUMBER 1:

For Section 2.4.6 Survey validity check”, the authors write Conducting surveys online expose researchers to more responders carelessness and fraud [36]. Even low levels of those can lead to a decrease in the quality of collected data. As recommended, six validated attention check items were added across the online surveys to detect careless or fraudulent responding. Four items were added to the first form and two to the second form (i.e. the form whose purpose is to test the temporal reliability of the tool), such as Please answer Strongly disagree” to this question” [37]. In this case, any other response than Strongly disagree” is considered incorrect. Correct answers received a score of zero and incorrect responses, a score of one. A total score out of 4 was computed for the first form and a total score out of 2 for the second form”.

When making use of Instructional Manipulation Checks (IMCs), one should acknowledge the limitations of including them. Oppenheimer et al. (2009) lists some of the disadvantages being that participants that are paying attention could feel insulted when running into an IMC and there might be a backlash by trying to foil the study and the concern that if an IMC is used to eliminate participants from the sample, then the external validity of the study could be harmed. 

References

Oppenheimer, D. M., Meyvis, T., & Davidenko, N. (2009). Instructional manipulation checks: Detecting satisficing to increase statistical power. Journal of Experimental Social Psychology, 45(4), 867-872. https://doi.org/10.1016/j.jesp.2009.03.009

Response: Indeed, IMCs are one type of attention checks commonly used by researchers. However, to control for careless responses, we chose to use another type of attention check item that is called instructed response items (Kung et al., 2018). We thank Reviewer 2 for this comment, as it carries its own risks, which we have now acknowledged and commented in the Discussion, as follows : 

Discussion

[…] Ultimately, conducting surveys online expose researchers to more responder’s carelessness and fraud [36]. Even though attention check items were added across the online surveys to control for careless or fraudulent responding, these measures carry risks. For instance, attention checks may influence responders’ further responses because of their very obvious caracter that inform responders about researchers’ intention to evaluate their attention level and put them in a more deliberative mindset [56]. Nevertheless, the incorporation of instructed response items has proven to ensure high quality data in previous research [37, 57]. Furthermore, Kung et al. [56] specifically investigated whether validity items influence participants’ responses and they found no evidence that they affect scale validity. Also, we can assume that the validity checks did not compromise the validity of the scale, but rather helped us make appropriate data quality decisions […]

This reference was added:

Kung, F. Y. H., Kwok, N., & Brown, D. J. (2018). Are Attention Check Questions a Threat to Scale Validity?: ATTENTION CHECKS AND SCALE VALIDITY. Applied Psychology, 67(2), 264‑283. https://doi.org/10.1111/apps.12108

MAJOR CONCERN NUMBER 2:

At the preliminary analysis, the authors write: From our initial sample (n = 612), we removed 132 participants who did not respond to the totality of the validity-check items, 3 respondents who completed the study twice on the online survey and 3 respondents who abandoned the completion of the form before the end. Regarding the validity-check, those who received a score > 1 were excluded, representing 9% of the total sample (n = 24).” 

This is all very unclear. The authors say 132 of 612 participants were removed. This is 21.6%  of the respondents (almost a quarter!). Then the authors say:

Of the 132: 

a) 3 respondents who completed the study twice on the online survey 

b) 3 respondents who abandoned the completion of the form before the end 

c) Regarding the validity-check, those who received a score > 1 were excluded, representing 9% of the total sample (n = 24)

But 3 + 3 + 24 = 30 (not 132). 

Response: We apologize for the confusion regarding the number of participants deleted. From the total sample, we removed 162 responders: 132 participants who did not respond to the totality of the validity check items + 3 respondents who completed the study twice on the online survey + 3 respondents who abandoned the completion of the form before the end + 24 respondents who did not pass the validity-check.. To facilitate understanding for future readers, we clarified this part in the Results section, as follows : 

Results

[…]From our initial sample (n = 612), we removed 162 participants for the following reasons: 132 participants did not respond to the totality of the validity-check items, 3 respondents completed the study twice on the online survey, 3 respondents abandoned the completion of the form before the end, and 24 respondents did not pass the validity check.

Regarding the latter, those who received a score > 1 were excluded, representing 9% of the total sample (n = 24). In the final sample (n = 450), only 3 values were identified as outlier data, representing 0.04% of total variables. All data from questionnaires had satisfactory skewness and kurtosis values, suggesting a normal distribution[…]

And there are problems with the choice of exclusion: 

For (a), how was it determined they completed the study twice? Using the computers IP address. This then would be wrong to do, as in a household, say the wife completed the questionnaire, then was so fascinated by / interested in it, she called her husband to complete the same questionnaire on the same computer.

Response: Regarding this criterion, we did not use the IP address only to determine that the same person completed the study twice. Indeed, for those excluded, the anonymity code and all demographic characteristics were also the exact same between the two completions (age, sex, educational level, occupation, health condition…). Based on all these similars information, we assumed that the same person completed the study twice and that it might compromise the validity of the study.

For (b), what if they completed 95% of the questionnaire and then left? I would then still advocate for using this data, as SPSS has ways of dealing with missing values (especially if the % missing values are as small as 5%). 

Response : As written above only 3 participants were excluded. We assume that adding 3 more participants will not change the results.

For (c), see my major concern number 1 of there being disadvantages to using IMCs that are not considered in the manuscript. 

Response : We have taken into consideration you major concern number 1. We thank again the Reviewer 2 for this point. 

MAJOR CONCERN NUMBER 3: 

The authors write, No normal distribution was considered when absolute values for skewness and kurtosis were greater than 3 and 10, respectively [38]”. One should use proper inferential statistical tests (with p-values) to test normality, such as the Kolmogorov-Smirnov test or the Shapiro-Wilk test (p > 0.05 indicated normality). One should never use skewness and kurtosis, as there are different recommendations in the literature about the cut-off scores, and there has no been no agreement on final” cut-off scores, i.e. one researcher might say the kurtosis cutoff is 10; someone else will say something else (there is no consensus on this in the literature). 

Response: We thank the Reviewer 2 for this comment. As mentioned in the main document, the final sample size for our study is 450. Empirical normality tests including Shapiro-Wilk or Kolmogorov-Smirnov tests cannot be used for samples larger than 300 because they are highly sensitive to the sample size. Besides, if the sample is > 300, the normality tests are likely to be significant, suggesting a violation of normality when this is not the case (Kim et al., 2012). To overcome this issue, the Skewness and Kurtosis appear to be a good alternative method to assessing normality in both small and large samples (Kim et al., 2013). Several authors recommend using the cutoffs of 3 and 10 for Skewness and Kurtosis (Brown, 2006; Kline, 2011; Wetson and Gore Jr, 2006).

Brown, T. A. (2006). Confirmatory factor analysis for applied research. Guilford Press.

Kline, R. B. (2011). Principles and practice of structural equation modeling (3rd ed.). Guilford Press.

Kim, H. Y. (2012). Statistical notes for clinical researchers: Assessing normal distribution (1). Restorative Dentistry. Endodontics, 37(4), 245–24. https://doi.org/10.5395%2Frde.2012.37.4.245

Kim, B., Kim, T., & Kim, J. (2013). Paper-and-pencil programming strategy toward computational thinking for non-majors: Design your solution. Educational Computing Research, 49(4), 437–459. https://doi.org/10.2190/ec.49.4.b

Weston, R., Gore, P.A., Jr., 2006. A brief guide to structural equation modeling. Couns.Psychol. 34 (5), 719–751. https://doi.org/10.1177/0011000006286345

Round 2

Reviewer 2 Report

The authors have addressed all my concerns adequately.